# Derivatives of Natural Chlorophylls as Agents for Antimicrobial Photodynamic Therapy

**DOI:** 10.3390/ijms22126392

**Published:** 2021-06-15

**Authors:** Nikita Suvorov, Viktor Pogorilyy, Ekaterina Diachkova, Yuri Vasil’ev, Andrey Mironov, Mikhail Grin

**Affiliations:** 1Department of Chemistry and Technology of Biologically Active Compounds, Medicinal and Organic Chemistry, Institute of Fine Chemical Technology, MIREA-Russian Technological University, 119571 Moscow, Russia; pogorilviktor@gmail.com (V.P.); mironov@mitht.ru (A.M.); michael_grin@mail.ru (M.G.); 2Department of Oral Surgery of Bororovsky Institute of Dentistry, II.M. Sechenov First Moscow State Medical University (Sechenov University), Trubetskaya St. bldg. 8\2, 119435 Moscow, Russia; 3Department of Operative Surgery and Topographic Anatomy, I.M. Sechenov First Moscow State Medical University (Sechenov University), Trubetskaya St. bldg. 8\2, 119435 Moscow, Russia; y_vasiliev@list.ru

**Keywords:** photodynamic therapy (PDT), photosensitizers (PS), antimicrobial PDT, chlorophyll, bacteriochlorophyll

## Abstract

The rapid growth of drug-resistant bacteria all over the world has given rise to a major research challenge, namely a search for alternative treatments to which bacteria will be unable to develop resistance. Photodynamic therapy is an approach of this kind. It involves the use of photosensitizers in combination with visible light at a certain wavelength to excite the former and generate reactive oxygen species. Various synthetic heterocyclic compounds are used as photosensitizers. Of these, derivatives of natural chlorophylls have a special place due to their properties. This review deals with the use of such compounds in antimicrobial PDT.

## 1. Introduction

The rapid growth of drug-resistant bacteria all over the world has given rise to a major challenge, namely a search for alternative treatments to which bacteria cannot develop resistance. Antibiotics widely used in the past half century are losing efficiency, and doctors increasingly adopt combined approaches to the treatment of infectious diseases.

Photodynamic therapy (PDT) involves the use of photosensitizers (PS) combined with light at a specific wavelength to excite the PS. The latter transfer energy to molecular oxygen. As a result, reactive oxygen species are formed, among which singlet oxygen is the main cytotoxic agent that can kill bacteria by oxidizing vital intracellular structures [1,2,3,4].

Antimicrobial PDT is a relatively new method for the treatment of bacterial infections. This method of fighting pathogenic microorganisms has a few advantages: the ability to cause localized impact on an infected site by means of local irradiation, along with high activity against Gram-positive and Gram-negative bacteria [5,6,7]. A photosensitizer penetrates the body and binds to a target cell, while causing no adverse effect by itself. Light is required to trigger photochemical reactions. Photosensitizers are capable of absorbing light in a strictly defined, narrow range, which is specific for each substance and is called the “absorption peak”. As a result, a transition of the dye molecule to an excited state occurs, followed by the generation of reactive oxygen species. Different photosensitizers have a different set of absorption bands, the position and extinction coefficient of which depend on the structure of the molecule. For example, for drugs of the chlorin group, the absorption maximum is 660–670 nm, and for bacteriochlorins it is 100 nm more. The shift of the absorption maximum to the near-IR region affects the depth of light penetration into the tissues. Thus, the average penetration depth of light is approximately 1–3 mm at 630 nm and up to 10 mm at 700–850 nm [8]. The presence of endogenous chromophores (hemoglobin, melanin, etc.) leads to the absorption of the energy of the incoming light, which weakens its penetration into tissues. Therefore, the greatest “transparency” of tissues is observed at wavelengths over 780 nm [9,10].

The antimicrobial PDT has several advantages over traditional antimicrobial chemotherapy with antibiotics: it offers a short inactivation time, does not lead to the bacterial resistance after multi-course treatment, does not cause dysbacteriosis, and is applicable to a wide range of target microorganisms [11]. A significant advantage of antimicrobial PDT is the possibility of a target effect in the infected areas located both on the surface and in the underlying tissues, without undesirable effects on the surrounding tissues and the natural microflora.

## 2. Principles of Antimicrobial PDT

It was shown in the 1990s that the susceptibility of Gram-positive and Gram-negative bacteria to PDT differs [12,13,14]. Neutral, anionic, and cationic PS molecules can efficiently bind to Gram-positive bacteria, whereas only cationic PS or neutral PS combined with membrane-permeating agents can bind to Gram-negative bacteria.

An Israeli team of scientists (Y. Nitan et al.) suggested the use of the polycationic drug Polymyxin B to increase the permeability of the outer membrane of Gram-negative bacteria and allow the PS to penetrate inside them [15]. A different approach was reported by G. Bertoloni et al. [16] who found that the use of ethylenediaminetetraacetic acid also increased the permeability of the outer membrane of Gram-negative bacteria, thus facilitating the internalization of the PS.

The difference in the sensitivity of bacteria to PDT is based on their structure. Gram-positive bacteria have a cytoplasmic membrane surrounded by a porous cell wall that consists of the peptidoglycan and lipoteichoic acids, which allow PS to readily bind to it. The shells of Gram-negative bacteria have the outer and inner cytoplasmic membranes separated by a layer of peptidoglycan. The outer membrane forms an efficient barrier to the permeation of many substances into a cell. The cell walls of pathogenic fungi have a thick layer of β-glucan and chitin, and they have an intermediate permeability between Gram-positive and Gram-negative bacteria. In contrast, for eukaryotic cells, the main components of membranes are neutral lipids, which leads to the absence of a charge on them [17].

Since the outer membranes of pathogenic bacteria and fungi carry a negative charge, molecules with positive charges are versatile broad-spectrum PS for antimicrobial therapy.

To date, many scientific works aimed at the use of synthetic porphyrins in the PDT have been published. Their cationic derivatives show bactericidal efficiency against Gram-positive and Gram-negative bacteria in several studies [18,19,20,21,22,23].

It is relevant to use antimicrobial PDT for destruction of bacterial biofilms. The latter are organized communities of bacteria integrated into a biofilm and protected from damage by the environmental factors [24,25]. The most important property of biofilms in clinical practice is their resistance to the antibacterial drugs which is 1000 times higher than for f bacteria in a planktonic culture [26]. The resistance is associated with the violation of the interaction between the antibiotic and its target. In planktonic bacteria, this phenomenon is observed at the level of a bacterial cell that is in direct contact with a drug. There are several classic mechanism types for these bacteria associations: a target modification, the antibiotic inactivation, an active antibiotic elimination from a microbial cell (‘efflux’), an impaired permeability of microbial cell’s external structures, and the formation of a metabolic “shunt”. The development of the biofilm resistance is more complicated. Mechanisms that prevent the action of antibiotics on pathogens into biofilms include [27]:Limited penetration of antimicrobial substances into biofilms, since the matrix slows down this process and binds antibiotics.Differences in metabolic activities and growth rates of individual bacterial cells.Restriction of nutrients and the altered microenvironment in the biofilm leading to a decrease in the rate of bacteria division and the targets number for the antibiotics action.Enhanced mutability of bacteria in the biofilm. Under the influence of strong antibiotic therapy, the rate of mutations leading to antibiotic resistance in *Pseudomonas aeruginosa* biofilms can reach 20% [24,25].Expression of undetected resistance genes.Interspecies transmission of genes for the antibiotic resistance, which is more successfully implemented in conditions of close contact of bacteria inside the biofilm.‘Efflux pump’. One of the mechanisms of resistance is the systems of the active release (‘efflux’) of the antibiotic, as well as the products of synthesis of the cell itself [28].Presence of bacterial cells capable of surviving under the stress conditions (persisted cells) in populations [25].

Obviously, as the bacterial resistance to the antibiotics increases, the efficiency of standard dosage schedules decreases. This stimulates the development of new antibacterial drugs. The main difference between photodynamic and antibiotic therapies is the lack of the resistance to singlet oxygen, that is the main cytotoxic agent. Focused irradiation of the lesion site leads to a local effect on the pathogen. Two mechanisms were suggested to explain the death of bacteria during PDT, namely, DNA damage and damage of the cytoplasmic membrane that results in the leakage of the cellular content into the external environment and malfunction of membrane transport systems [29,30,31,32,33,34,35].

To date, numerous PS have been synthesized for antimicrobial PDT, both based on the synthetic tetrapyrrole compounds and other dyes (acridine, toluidine, hypericin, etc.) [36,37,38,39,40,41,42]. Ammonium or pyridinium groups are incorporated into such pigments to give them greater antimicrobial activity due to a positive charge on the nitrogen atom. Recently, a number of review articles have been published on the subject of antimicrobial photodynamic therapy [43,44,45,46,47,48,49]. Our review focuses exclusively on the application of natural chlorin derivatives.

## 3. Neutral and Anionic Chlorin Derivatives for Antimicrobial PDT

In contrast to synthetic dyes, the natural chlorin derivatives have several advantages. The main sources of such compounds include plants and the algae containing chlorophylls *a* and *b* (Figure 1). Such raw materials are readily available and inexpensive, and methods for extracting pigments from them do not require large labor costs. Due to the reduced pyrrole ring, compounds **1** and **2** and their derivatives manifest absorption in the near infrared region with high molar extinction values, which makes it possible to affect the deep zones. Bacteriochlorophylls that are also abundant in the nature are an independent group of natural chlorophylls [50,51]. Bacteriochlorophylls *a–g* that differ both in the degree of macrocycle hydrogenation and in substituents on its periphery are known to date. The main natural sources of these compounds include purple bacteria, green sulfur bacteria, and heliobacteria. Bacteriochlorophyll *a* **3** deserves a special attention, as its absorption maximum is shifted even more strongly to the red region than for the chlorophyll *a*. It is obtained from the biomass of purple bacteria such as *Rhodobacter*
*sphaeroides*, *Rhodobacter*
*roseapersiana,* and *Rhodobacter*
*capsulata*. The spectral properties of the aforementioned compounds are affected by the conjugation length in the macrocycle, which is violated during the reduction of pyrrole rings. Reduction of one or both double bonds does not affect the aromaticity of the resulting chlorins and bacteriochlorins, and a change in symmetry (the appearance of sp^3^-hybrid carbon atoms) leads to a significant batachromic shift of the long-wavelength absorption band [52]. In the series of the porphyrins, chlorins, and bacteriochlorins, the chemical and photo stabilities decrease, since the reduced pyrrole rings are prone to the oxidation.

They produce only one type of bacteriochlorophyll, thus the isolation and purification of the pigment become much easier [53,54,55]. Derivatives of the natural chlorophyll and bacteriochlorophyll, unlike their synthetic analogs, have low dark toxicity and are readily excreted from the body as harmless metabolites. Today, these compounds are widely used in the antitumor PDT [56,57,58,59,60]. However, due to the properties mentioned above, they have high applied potential in other areas of medicine. This review deals with the use of derivatives of natural chlorophylls in the antimicrobial PDT.

K. da Silva Souza Campanholi et al. obtained an extract of chlorophylls *a* and *b* from *Tetragoniatetragonoides*. Pigments **1** and **2** were isolated from plant materials by extraction with an ethanol-water mixture and then converted to a micellar form using Pluronic F127, a block copolymer of polyethylene glycol and polypropylene glycol. The authors estimated the photoinduced cytotoxicity of the specimen thus obtained against *Staphylococcus aureus* bacteria cells. The tests showed that complete inactivation of bacterial cells occurred, while no significant cytotoxic effects were observed in a similar experiment with healthy cells. Thus, it was shown that even unmodified chlorophyll *a* and chlorophyll *b* exhibited high antimicrobial activity under irradiation [61].

In recent years, PDT has become relevant for solving a number of problems related to food safety, including improper storage conditions [62]. The concept of active packaging is becoming an alternative to conventional packaging, allowing to reduce the growth rate of the dangerous microorganisms due to the addition of various substances to the main composition [63,64]. In the work of Rizzi et al., various hydrophobic composite films based on chitosan and chlorophyll *a* were obtained [65]. The authors showed that the latter has a strong affinity for chitosan due to the interaction with its amino groups. When such films were irradiated with visible light, the generation of singlet oxygen was observed, which was detected by luminescence infrared spectroscopy, as well as by using chemical traps. According to the authors, the proposed composite material can be successfully used to extend the shelf life of food products.

Gerola et al. investigated the photo-physical properties (photostability, singlet oxygen quantum yield) and the antimicrobial effect of chlorophyll derivatives **4 a**–**f** (Figure 2) [66]. Pheophytins **4 a**–**d** did not aggregate at high concentrations in micellar systems (Tween 80 or P- 123). Photostability changed as follows: **4c** > **4d≈4e** > **4b≈4f** > **4a**. Derivatives **4b** and **4e** in an aqueous solution of Tween 80 showed the highest quantum singlet oxygen yield compared to other derivatives. The antimicrobial effect of pigments **4a**–**f** was investigated against *Staphylococcus aureus*, *Escherichia coli*, *Candida albicans*, and *Artemia salina*. Metal-free pheophytin **4e** showed the best results against all tested microorganisms, pigment **4f** had the highest dark cytotoxicity, and the Cu-complex of pheophytin **4c** did not show photodynamic inactivation effect. The authors showed that chlorophyll *a* **4a**, pheophytin zinc complex **4b**, pheophytin **4d**, and pheophorbide **4e** are promising candidates for antimicrobial PDT.

F. Ayaz and co-authors investigated the antimicrobial, anti-inflammatory, and immunostimulant properties of metal complexes of pyro pheophorbide methyl ester [67]. Chlorophyll derivatives are known for their ability to regulate the immune system [68,69]. The authors investigated Cu, Ni, Mg and Zn metal complexes, as well as pyro pheophorbide in the form of a free base (Figure 2). Earlier, it was shown that these metals play an important role in the activation of cells of the immune system, and consequently in the generation of an immune response to stimulating signals [70,71,72]. This stimulating signal can be tumor cells or infectious agents such as bacteria and viruses. It was shown that compounds **4g**–**k** had an immunostimulatory effect on mammalian macrophages. This effect was determined based on the levels of the anti-inflammatory cytokines’ secretion. Moreover, after photoactivation, the immunostimulant activity decreased for all compounds, except for **4g** and **4h**. To assess the immunomodulatory effect, including the regulation of the immune system cells in the presence of stimulating signals [73,74], the authors used stimulant mimics as the lipopolysaccharide and lipoteichoic acids. It was found that compounds **4i**–**k** have the significant photoinduced immunomodulatory activity. This effect combined with the photodynamic potential of the compounds studied can be used in the treatment of the infectious diseases.

Photosensitizers of the chlorin series employed in clinical antitumor PDT were also used to inactivate bacteria. For example, Ulatowska-Jarza et al. studied the antimicrobial activity of the Photolon, which contains salts of chlorin e_6_ obtained from natural chlorophyll *a*. The study was carried out with *Escherichia coli* strains obtained from poultry and cows. Bacteria were incubated with a Photolon solution and then exposed to laser light with a wavelength of 662 nm. Additionally, similar studies were carried out with a photosensitizer in a gel applicator. It was demonstrated that both dosage forms of Photolon inhibited the growth of bacteria under irradiation, while no toxic effect was observed in the control groups. According to the authors, the use of this sensitizer in an applicator form can be used both for interstitial therapy and for disinfection of various liquid media [75].

Photodynamic therapy (PDT) was used by M.T. Garcia and A.H.C. Pereira for the inactivation of oral pathogens by different photosensitizers [76]. This study evaluated the efficacy of PDT against *Streptococcus mutans* biofilms using two second-generation PS, Photoditazine^®^ (PDZ) and Fotoenticine^®^ (FTC), which are anionic derivatives of chlorin e_6_ with specific counter-ions. These PS were compared to methylene blue (MB), a dye with a proven antimicrobial activity against *Streptococcus mutans*. Suspensions of *S. mutans* were cultured in contact with bovine tooth disks for biofilm formation. After 48 h, the biofilms were treated with PDZ (0.6 mg/mL), FTC (0.6 mg/mL) or MB (1 mg/mL) and submitted to the laser irradiation (660 nm, 50 mW/sm^2^). The biofilms were quantified by the determination of CFU/mL count and analyzed by scanning electron microscopy (SEM). All PS used for PDT reduced the number of *Streptococcus mutans*, with a statistically significant difference compared to the untreated groups. PDT achieved microbial reductions of 4 log with MB and 6 log with PDZ, while the use of FTC resulted in the complete elimination of *Streptococcus mutans* biofilms. SEM analysis confirmed the CFU/mL results, showing that all PS, particularly FTC, were able to detach the biofilms and to eliminate the bacteria. Authors showed that PDT mediated by chlorin-type PS has exhibited the greater antimicrobial activity against *Streptococcus mutans* than MB-mediated PDT. This fact indicated the use of these PS c for the dental caries control.

Research team from Republic of Korea has demonstrated that the antimicrobial PDT using chlorin e_6_ **5** combined with halogen light could be an alternative therapy for acne vulgaris [77,78]. Halogen lights, having approximately 35% of their light emitted falling in the 600–900 nm range, emit spectra of light that are very similar to sunlight. *Propionibacterium acnes* is a Gram-positive anaerobe normally located in the human sebaceous glands and has been associated with the inflammatory phase in acne vulgaris [79]. The results obtained by the authors reveal that chlorin e_6_-mediated PDT showed strong antimicrobial activity against skin bacteria, including *P. acnes* in vitro, as well as therapeutic potential against *Propionibacterium acnes* induced inflammation *in vivo*. The growth of *Propionibacterium acnes* (3 strains) and *Staphylococcus aureus* were completely inactivated by chlorin e_6_ **5** with and without irradiation even at a low dose, showing stronger antibacterial capacity than 5-Aminolevulinic acid. The superior antimicrobial activity is due to the generated free radicals from activated chlorin e_6_ upon illuminating halogen light.

## 4. Antimicrobial Cationic Derivatives of Natural Chlorins

Molecules comprising positively charged groups are the most efficient photosensitizers for the antimicrobial PDT. They enable a more efficient binding of a PS to the cell walls of both Gram-positive and Gram-negative bacteria. M. Hamblin et al. obtained a polycationic PS [80] based on a conjugate of chlorin e_6_ with poly-L-lysine **7**. Polycationic, neutral, and anionic forms of this molecule were obtained to study the charge effects. Chlorin e_6_ **5** and chlorin e_6_ aminoethylamide **6** were taken as the reference PS. The tests were carried out with *Porphyromonas gingivalis* (Gram-negative bacteria) and *Actinomyces viscosus* (Gram-positive bacteria). HCPC-1 epithelial cells were taken as the reference.

The research results showed that polycationic conjugate **7** bound well with both Gram-positive and Gram-negative bacteria, showing high phototoxicity when irradiated with red-light. Compared with polycationic chlorin **7**, the compounds **5** and **6** accumulated in bacterial cells 1.3–2 times worse. In this case, the cationic photosensitizer **7** bound to epithelial cells 20–100 times weaker than bacterial ones, which allows us to conclude that it is selective to bacterial pathogens, and consequently safe for the use in antimicrobial PDT.

M. Hamblin et al. studied the efficiency of cationic PS with yet another polymer, polyethyleneimine [81]. Conjugates of chlorin e_6_ with polyethyleneimines having various structures were obtained (**8**,**9**).

The tests were performed with Gram-positive and Gram-negative bacteria, as well as with yeast microorganisms under red-light irradiation. Conjugate **9**, which has a large molecular mass and a branched structure, showed good results with all types of pathogens used, including Gram-negative *P. aeruginosa* bacteria that are very resistant to external impacts, whereas linear conjugate **8** proved to be efficient against Gram-positive bacteria and yeast only. In contrast to polylysine conjugates, conjugates **8** and **9** are resistant to degradation by proteases, for example, trypsin hydrolyzes lysine–lysine peptide bonds, which gives them an advantage when used in in vivo PDT.

Y. Wang et al. reported on the synthesis of a chlorin e_6_ conjugate with a cationic peptide obtained from a transcriptional transactivator (TAT) and belongs to the class of cell-penetrating peptides [82]. Due to the large number of cationic centers in the TAT peptide, it is able to interact purposefully with the negatively charged bacterial cell wall with subsequent internalization through endocytosis [83,84]. The resulting conjugate **10** was used to prepare self-organizing nanoparticles loaded with the tinidazole antibiotic employed in the clinics to achieve a synergistic anti periodontitis effect by combining PDT with the antibacterial therapy. Compared to the free chlorin e_6_, the nanoparticles have a significantly better ability to penetrate the cells of periodontal pathogenic bacteria and provide an enhanced efficiency of PDT toward both periodontal pathogenic bacteria and monocyte macrophages. On irradiation with light at a wavelength of 635 nm, the antibiotic-loaded nanoparticles exhibited an efficient combined antimicrobial effect [85].

Kustov A. V. et al. studyied the effect of the structure of chlorinated pigments **11**–**13** containing from one to three positively charged groups on the spectral, photophysical and antimicrobial properties of the latter [86]. The authors used cationic chlorins obtained from methyl pheophorbide by opening the exocycle with diaminoethane or methylamine, as well as aminomethylation of the vinyl group with bis-(*N*, *N*-dimethylamino)-methane, as the photosensitizers under study. At the second stage of the synthesis, the target amines were alkylated with methyl iodide to obtain cationic forms. The pigments **11**–**13** described in the work turned out to be highly soluble in water and stable both in solution and in crystalline form. They generate singlet oxygen with good quantum yield (0.53–0.65 depending on the solvent) and exhibit an affinity for lipid cell membranes. The following opportunistic microorganisms (Gram-positive bacterium *Staphylococcus aureus* (6538-P ATCC = 209-P 295 FDA strain), Gram-negative bacterium *Escherichia coli* (M-17 strain), and yeast-like fungi *Candida albicans* (CCM 8261 ATCC 90028 strain)) were selected to study antimicrobial PDT.

Microbiological studies have shown that even highly diluted solutions of pigments effectively kill the Gram-positive bacteria and fungi, but do not act on the Gram-negative bacterial cells. An increase in the concentration of photosensitizers **11**–**13** and the addition of certain agents, including ethylenediaminetetraacetic acid calcium disodium salt, or Tween 80 enhance the permeability of the outer membrane of Gram-negative bacteria, provide complete inactivation of opportunistic microflora. These results show the great potential of charged chlorin pigments for the treatment of wound infections and bacterial biofilms, and the positively triply charged pigment was most effective both in the absence and upon irradiation with light. The selectivity of the method of antimicrobial photodynamic therapy includes focused light irradiation of the microbial lesion area. Therefore, the cytotoxicity of PS in the absence of light (dark toxicity) should be minimized.

In another article, the same group of authors synthesized two new chlorin-type photosensitizers, one of which contains a covalently attached myristic acid residue (**14**)-to increase the tropism to the membrane, and another PS **15** modified with myristic acid additionally contains two cationic trimethylammonium groups-to enhance hydrophilic properties and increase affinity for Gram-negative bacteria [87]. To dissolve the obtained PSs, the authors used the Tween 80 non-ionic surfactant. The investigation of their antimicrobial activity in vitro has indicated that PSs are able to kill both archival opportunistic microorganisms and Gram-negative hospital-acquired bacteria. The cationic chlorin has also demonstrated efficient photoinactivation of antibiotic resistant strains of *Pseudomonas aeroginosa* and *Escherichia coli* in the animal model of a wound infection.

The archival opportunistic pathogens were presented by Gram-positive bacteria *Staphylococcus aureus* (6538-P ATCC = 209-P FDA strain), Gram-negative bacteria *Escherichia coli* (M-17 strain) and yeast-like fungi *Candida albicans* (CCM 8261 ATCC 90028 strain). The authors showed that “Fotolon” being the molecular complex of anionic chlorin e_6_ with its carrier polyvinyl pyrrolidone [88] is not able to eradicate Gram-negative bacteria.

M. Grin and co-workers obtained a photosensitizer based on chlorin e_6_ aminobutyl amide (**16**) and its cationic derivative (**17**) (Figure 3) [89]. The antibacterial activity of PS **16** and **17** was studied by the serial dilution method in a liquid medium for the inhibition of bacterial reproduction. The Gram-positive *Staphylococcus aureus 209P* bacteria and the Gram-negative *Pseudomonas aeruginosa PAO1* bacteria were used in the tests.

It was found that compounds **16** and **17** had a phototoxic effect toward Gram-positive *S. aureus 209P* bacteria leading to 100% inhibition of bacterial growth at a minimum inhibiting concentration (MIC) of 160 nM with respect to the active compound. At the same time, PS **16** and **17** did not inhibit the growth of Gram-negative bacteria *Pseudomonas aeruginosa* in the entire concentration range studied (up to 5 μM).

To develop this work further, cationic photosensitizers with heterocyclic moieties were obtained (Figure 3) [90,91]. Derivatives of natural chlorins and bacteriochlorins containing a nicotinic acid residue were selected for the synthesis of mono cationic PS. To create a cationic moiety, quaternization of the nitrogen atom in the pyridine ring involving reactions with methyl iodide and propyl iodide to give compounds **18** and **19** was used.

The effect of alkyl groups at the nitrogen atom of the pyridine ring on the photoinduced efficiency of cationic purpurinimides **18** and **19** was estimated based on the decrease in the titer of bacteria *Pseudomonas aeruginosa* in a clinical isolate after the biofilm was destroyed by sonication. Petri dishes with bacteria treated with a photosensitizer were irradiated with red light from a xenon lamp with a mean power density of 30 ± 1 mW/sm^2^ and a light dose density of D = 36 J/sm^2^ (within Δt = 20 min).

The dependence of the efficiency of photodynamic inactivation on the radiation dose was studied for purpurinimides **18** and **19** in a cremophoric emulsion at 1 mM concentration of each of the PS. Complete inactivation was observed in the presence of compound **18** at a light dose density of D = 105 J/sm^2^. However, prolonged (60 min) irradiation in the presence of compound **19** or in the absence of a photodynamic agent gave no observable death of bacteria in biofilms. It was found that in biofilms incubated for 1 h without irradiation with a 1 mM solution of photosensitizer **17**, partial inactivation of bacteria was observed (by two orders of magnitude compared to the control without a photosensitizer). This may apparently be due to the sonodynamic excitation of the photosensitizer during sonication that was performed to destroy biofilms for counting viable colonies [92,93,94].

Studies have shown that the suggested cationic photosensitizer for antibacterial photodynamic therapy based on the methyl ester of 13^3^-*N*-(*N*-methylnicotinyl) purpurinimide **18** has high antibacterial efficiency against biofilms, in contrast to propyl derivative **19**.

The photodynamic therapy method using cationic purpurinimide **18**, which absorbs in the region of 700 nm, has a drawback since if there is considerable invasion of a chronic infectious damage into the underlying tissues, a considerable fraction of light is absorbed by endogenous chromophores and, accordingly, does not excite the photosensitizer molecules. As a result, the efficiency of photodynamic treatment decreases. Bacteriochlorophyll derivatives are most promising for PDT due to their absorption in the near infrared spectral region where light penetrates tissue to a greater depth and provides a higher therapy efficiency. Therefore, Grin et al. obtained a similar cationic PS based on bacteriochlorophyll *a,* and its pharmaceutical form.

To create an experimental wound infection, the mice were inflicted with surgical wounds, followed by infection with bacterial suspensions of *Pseudomonas aeruginosa* or *Staphylococcus aureus*. After infection of the wounds, the animals were once intravenously injected with the tested dosage form at various doses and irradiated with light with a wavelength of 822 nm. As a result of the studies, the minimum effective dose of PS **20** and **21** was determined, amounting to 20 mg/kg and 10 mg/kg for *Pseudomonas aeruginosa* and *Staphylococcus aureus*, respectively [95].

M. Grin and co-workers also developed and synthesized a polycationic photosensitizer **21** containing several *N*-methylpyridinium residues [96]. For this purpose, derivatives of natural chlorophylls were obtained with cyclene, which was modified with nicotinic acid residues followed by alkylation of nitrogen atoms for localizing the positive charge at the periphery of the chlorin macrocycle. A study of the photoinduced antimicrobial activity of PS **21** showed that only Gram-positive bacteria *Staphylococcus aureus* were inhibited at a MIC of 2.5 μM, whereas no photoinduced activity against Gram-negative *Escherichia coli* bacteria was observed. Thus, an increase in the number of charged nicotinyl groups in the chlorin molecule reduces the photoinduced bacteriostatic effect toward Gram-negative bacteria, apparently due to steric factors that complicate the interaction of a PS with the bacterial membranes. Therefore, compound **21** was found to be less effective than the above-described mono cationic derivative **18**.

To study the effect of the number of the cationic centers and their positions on the periphery of the chlorin macrocycle, we are currently working on the synthesis of pyridazine-substituted derivatives of chlorins e_6_ and p_6_ (unpublished data). Inverse electron demand Diels–Alder reactions are used to modify the pigments. It was previously shown that this reaction occurred in high yields and had a few advantages over other methods for modifying chlorophyll *a* derivative [97,98]. We are currently planning to study the photoinduced toxicity of the PS obtained against Gram-positive and Gram-negative bacteria to identify the structure–activity relationship.

## 5. Conclusions

Antimicrobial drugs play an important role in modern medicine. Their use has led to a decrease in mortality rate and an increase in the life expectancy, and their importance in the chemotherapy of bacterial infections is currently difficult to overestimate.

However, resistance to available antibiotics in pathogenic bacteria is a global problem, as the number of multidrug-resistant strains increases dramatically every year and spreads around the world.

Multidrug-resistant pathogens, especially ESKAPE bacteria (Enterococcus faecium, Staphylococcus aureus, Klebsiella pneumoniae, Acinetobacter baumannii, Pseudomonas aeruginosa, Enterobacter spp.) as well as pathogens of tuberculosis (Mycobacterium tuberculosis), can acquire drug resistance with various mechanisms of chemical resistance. The use of antibiotics in high doses leads to side properties. Thus, the search for new ways to overcome antibiotic resistance is an urgent and important task of modern medicinal chemistry.

The known structures of molecules of natural chlorophylls, including chlorophyll *a*, and bacteriochlorophyll *a*, open up wide possibilities for their directed modification in order to obtain highly stable derivatives with improved photophysical characteristics, increased hydrophilicity, as well as to create conjugates with other molecules. Introduction to the periphery of the chlorin macrocycle of heterocyclic fragments, including pyridine derivatives, azamacroheterocycles, as well as aminoalkyl groups and transition metal cations into the inner spheres of subunits, allows for the targeted synthesis of pharmacologically active compounds for various biomedical applications.

A new way of influencing microbial pathogens is photodynamic inactivation with the help of singlet oxygen generated by PS irradiation. The most resistant to PDT are Gram-negative bacteria, which is associated with the low permeability of their outer membrane for dyes. The negative charge of the outer surface of bacterial cells determines the active binding with them and, accordingly, the pronounced antibacterial activity of cationic dyes.

Optimization of the structure of chlorin systems with one or more positive charges, absorbing light in the region of 700–800 nm, makes it possible to increase the efficiency of photodynamic therapy of local foci of infection by increasing the proportion of light radiation that excites photosensitizer molecules in tissues.

Assessing the current state of development of the method of antimicrobial photodynamic therapy, it should be noted that the level of fundamental development, and especially practical implementation, still lags far behind PDT of cancer. However, given the growing pace of research in this area, in the coming years we can expect the formation of a new and effective way to combat infections of superficial localizations. This review presents the main directions of the modification of natural chlorins as a platform for the creation of photosensitizers for antimicrobial PDT.

## Figures and Tables

**Figure 1 ijms-22-06392-f001:**
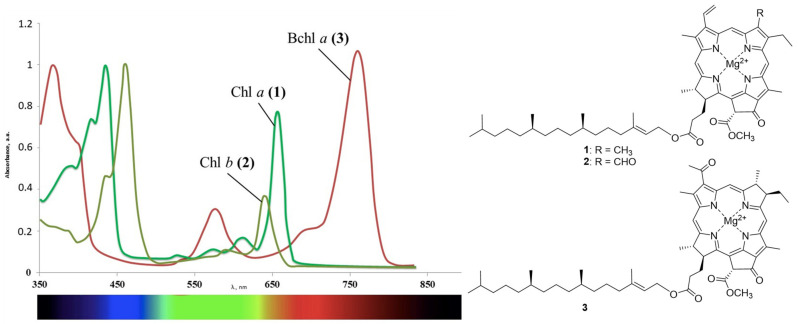
Chlorophyll *a*
**(1)**, chlorophyll *b*
**(2)**, bacteriochlorophyll *a*
**(3)**, and their absorption spectra.

**Figure 2 ijms-22-06392-f002:**
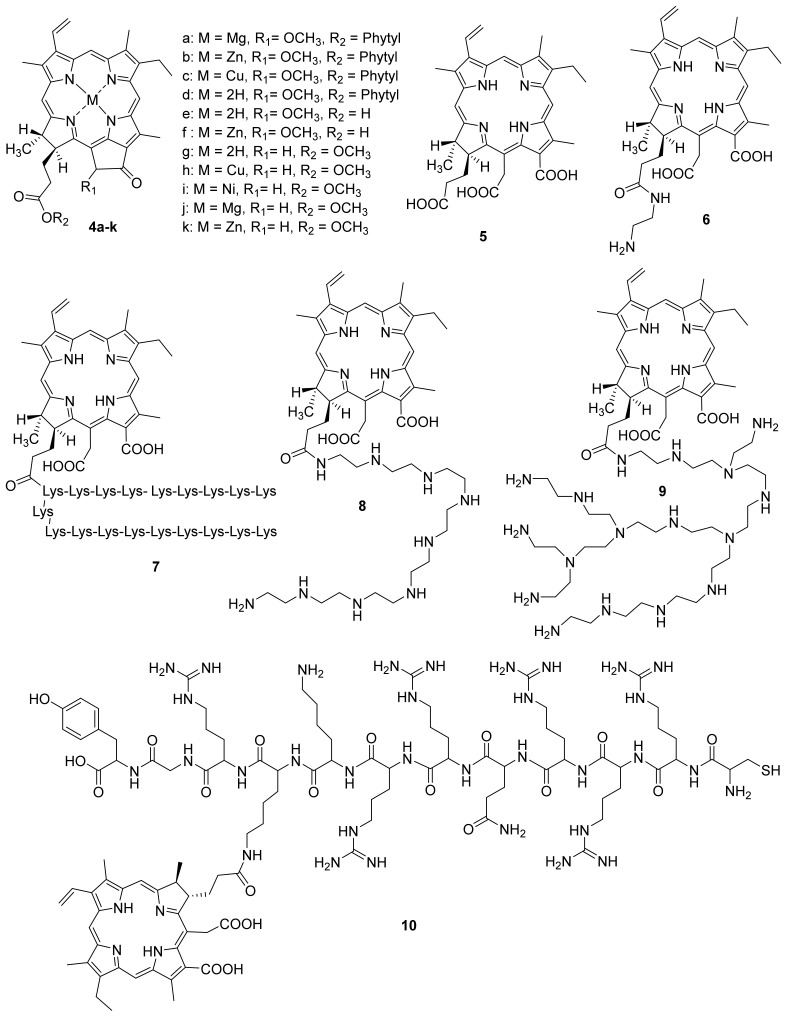
Chlorophyll *a* derivatives used in antimicrobial PDT.

**Figure 3 ijms-22-06392-f003:**
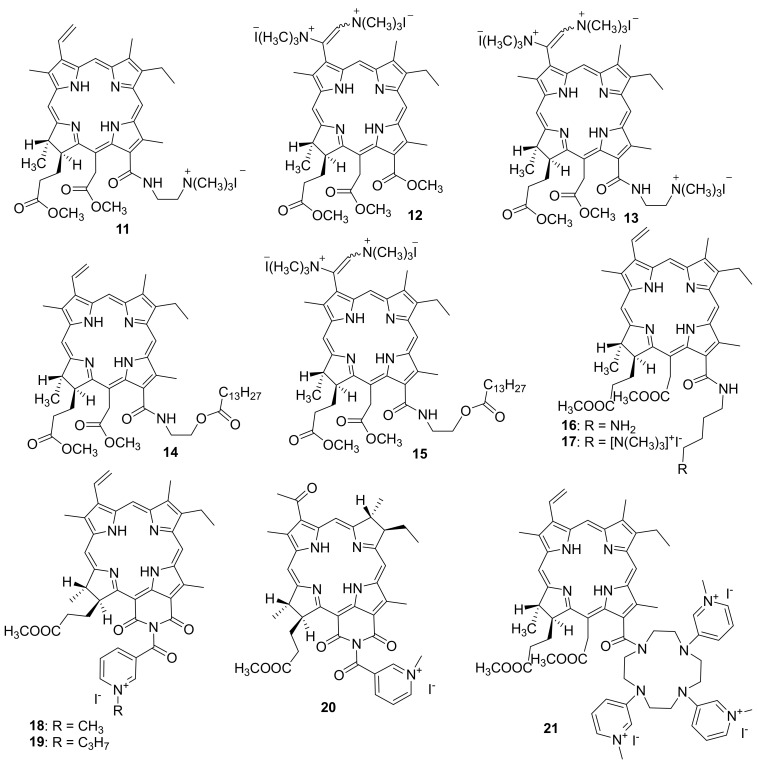
Cationic derivatives based on chlorins and heterocyclic compounds.

## Data Availability

Not applicable.

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
