# Peer review of "Derivatives of Natural Chlorophylls as Agents for Antimicrobial Photodynamic Therapy"

_ijms, 2021, doi:10.3390/ijms22126392_

Round 1
Reviewer 1 Report
- page 2-3, line 90-120 - maybe Author should provide the scheme/figure of molecular interactions,
- fig. 1-3 should be better quality
Author Response
The team of authors would like to thank the referee for the analysis of the peer-reviewed work and the comments made, which helped to improve the quality of the article.
1) page 2-3, line 90-120 - maybe Author should provide the scheme/figure of molecular interactions
Since the focus of the review is on antimicrobial PDT, in our opinion it would be unnecessary to present a figure with a detailed mechanism. However, this paragraph has been shortened and optimized for better reader convenience.
2) fig. 1-3 should be better quality
The quality of the figures has been improved.
Reviewer 2 Report
The present paper by N. Suvorov et al. is a brief review on natural chlorophyll derivatives in photodynamic therapy (PDT), with a focus on the treatment of microbial infections (antimicrobial PDT). As the authors state, photodynamic therapy represents a promising alternative in the treatment of infections, particularly because, unlike with classic antibiotic treatment, no resistance to reactive oxygen species can be developed by pathogens. The focus is here on chlorophyll derivatives, a highly important class of photosensitizers (PS), in particular because of their abundance in nature and of their structural singularity and versatility providing unique optical properties. To the best of my knowledge, the last review dedicated to all tetrapyrrole derivatives in PDT dates back from 2017, and does not focus particularly on antimicrobial applications, while another review from 2020 focuses on bacteriochlorins specifically. The present review is therefore timely and in a hot contemporary research area. However, there are a number of important points that need to be addressed in this review, and I can recommend publication of this manuscript as it is.
The review is 10 pages long, and provides 48 references. It is globally under-referenced and sometimes improperly referenced (section 4 in particular, see comments below). Photodynamic therapy is a very large field and one cannot expect a comprehensive overview, however it would be preferable to include more background information, and/or to refer the reader to additional recent reviews. A few examples of relevant recent reviews is listed below:
- Recent progress in photosensitizers for overcoming the challenges of photodynamic therapy: from molecular design to application, Zhao, Xueze; Liu, Jiangping; Fan, Jiangli; Chao, Hui; Peng, Xiaojun, Chemical Society Reviews (2021), 50(6), 4185-4219.
- Phototherapy-​based combination strategies for bacterial infection treatment, Wei, Guoqing; Yang, Guang; Wang, Yi; Jiang, Hezhong; Fu, Yiyong; Yue, Guang; Ju, Rong, Theranostics (2020), 10(26), 12241-12262.
- Design of photosensitizing agents for targeted antimicrobial photodynamic therapy, Klausen, Maxime; Ucuncu, Muhammed; Bradley, Mark, Molecules (2020), 25(22), 5239.
- Bacteriochlorins and their metal complexes as NIR-​absorbing photosensitizers: properties, mechanisms, and applications, Pucelik, Barbara; Sulek, Adam; Dabrowski, Janusz M., Coordination Chemistry Reviews (2020)
- Design features for optimization of tetrapyrrole macrocycles as antimicrobial and anticancer photosensitizers, Martinez De Pinillos Bayona, Alejandra; Mroz, Pawel; Thunshelle, Connor; Hamblin, Michael R., Chemical Biology & Drug Design (2017), 89(2), 192-206.
- Photodynamic therapy: a new antimicrobial approach to infectious disease?, Hamblin, Michael R.; Hasan, Tayyaba, Photochemical & Photobiological Sciences (2004), 3(5), 436-450.
- New photosensitizers for photodynamic therapy, Abrahamse, Heidi; Hamblin, Michael R., Biochemical Journal (2016), 473(4), 347-364.
In this sense, section 4 of the article is also very heavy on the work of the present authors, including comments about unpublished results and long experimental details that do not necessarily serve the discussion. Concerningly, this section is also very badly referenced as I was unable to find where most of the results presented in it came from. In contrast, the authors omitted other significant recent papers on chlorophyll/chlorin derivatives in aPDT such as (non-exhaustive list):
- Antimicrobial photodynamic therapy using chlorin e6 with halogen light for acne bacteria-​induced inflammation, Jeon, Yu-Mi; Lee, Hwan-Suk; Jeong, Dongjun; Oh, Hae-Keun; Ra, Kyu-Hwan; Lee, Mi-Young, Life Sciences (2015), 124, 56-63.
- Molecular interactions, characterization and photoactivity of Chlorophyll a​/chitosan​/2-​HP-​β-​cyclodextrin composite films as functional and active surfaces for ROS production, Rizzi, Vito; Fini, Paola; Fanelli, Fiorenza; Placido, Tiziana; Semeraro, Paola; Sibillano, Teresa; Fraix, Aurore; Sortino, Salvatore; Agostiano, Angela; Giannini, Cinzia; et al, Food Hydrocolloids (2016), 58, 98-112.
- Synthesis and investigation of water-soluble chlorophyll pigments for antimicrobial photodynamic therapy, Kustov, Andrey V.; Belykh, Dmitriy V.; Smirnova, Nataliya L.; Venediktov, Evgeniy A.; Kudayarova, Tatyana V.; Kruchin, Sergey O.; Khudyaeva, Irina S.; Berezin, Dmitriy B., Dyes and Pigments (2018), 149, 553-559.
- Synthesis and investigation of novel chlorin sensitizers containing the myristic acid residue for antimicrobial photodynamic therapy, Kustov, Andrey V.; Kustova, Tatyana V.; Belykh, Dmitry V.; Khudyaeva, Irina S.; Berezin, Dmitry B., Dyes and Pigments (2020), 173, 107948.
- Effects of metal and the phytyl chain on chlorophyll derivatives: physicochemical evaluation for photodynamic inactivation of microorganisms, Gerola, Adriana P.; Santana, Amanda; Franca, Polyana B.; Tsubone, Tayana M.; de Oliveira, Hueder P. M.; Caetano, Wilker; Kimura, Elza; Hioka, Noboru, Photochemistry and Photobiology (2011), 87(4), 884-894.
- Bacterial infection microenvironment sensitive prodrug micelles with enhanced photodynamic activities for infection control, Wang, Shuting; Fang, Yu; Zhang, Zequn; Jin, Qiao; Ji, Jian, Colloid and Interface Science Communications (2021), 40, 100354.
- Photodynamic therapy mediated by chlorin-type photosensitizers against Streptococcus mutans biofilms, Terra Garcia, Maira; Correia Pereira, Andre Henrique; Figueiredo-Godoi, Livia Mara Alves; Jorge, Antonio Olavo Cardoso; Strixino, Juliana Ferreira; Junqueira, Juliana Campos, Photodiagnosis and Photodynamic Therapy (2018), 24, 256-261.
Overall, the review slightly lacks scientific rigour in referencing, and insight in the way reports are presented. The bigger picture is often slightly omitted, with reports presented too independently from each other, and with a concluding section falling outside the point of the paper. In details, the following points need to be addressed:
Page 1, lines 37-42, the authors write “Photosensitizers are capable of absorbing radiation in the entire visible region of the spectrum”. This is misleading at best, inaccurate at worst. Photosensitizers will absorb light in their absorbance peaks, shining light of other wavelengths cannot result in promotion to the excited state and generation of reactive oxygen species. It might be a shortcut due to the fact that chlorophyll derivatives do have several absorption peaks across the visible, but it remains untrue as a general information. Please rectify.
Page 2, lines 51-54, the authors write “it was shown in the 1990s that the susceptibility of gram-positive and gram-negative bacteria to PDT differs.” This sentence is not referenced; the next referenced cited is from 2002 and deals with porphycene–polylysine conjugates for both types of bacteria.
Page 2, line 69-71, since the authors are talking about a “passive targeting” based on electrostatic interactions between cationic PS and pathogens, it would be beneficial to discuss briefly about the selectivity of this strategy between pathogens and mammalian cells.
Page 2, line 72-74, the paragraph is under-referenced.
Page 2, line 75-80, this paragraph is not referenced, references 12 and 13 should be cited earlier.
Page 2, line 82, “many times higher” is vague, giving an order of magnitude would benefit the discussion.
Page 3, line 123-124, the authors write “The main difference between photodynamic therapy and antibiotic therapy is the lack of resistance to singlet oxygen”. Although the difference is true, the sentence lacks context, as photodynamic therapy requires the use of light and is therefore a localised treatment, while antibiotic therapy can be applied in the whole body. Reformulation is advised.
Page 3, line 138, it would be beneficial to the reader to explain why the reduced pyrrole ring is responsible for a near-infrared absorption with high extinction coefficient in chlorophyll derivatives. Same goes for bacteriochlorophyll derivative 3 in which the effect is even more pronounced. As a more general comment, this section would benefit from having a brief introduction to the different classes of tetrapyrrole macrocycles encountered in the manuscript (porphyrin, chlorin, bacteriochlorin) with comments on the effect of hydrogenation/symmetry on their absorption properties. This would also be the opportunity to discuss a point that is entirely absent from the paper, i.e. the stability of bacteriochlorin derivatives.
Page 4, line 152, the sentence “Today these compounds are widely used in antitumor PDT.” is not referenced.
Page 4, Figure 1, the axes on the absorption spectrum do not have any title or units.
Page 5, compounds 5c-e by Ayaz et al., shown on figure 2, are not discussed in the paragraph (lines 182-192). In addition, the paragraph focuses on the fact that compounds 5a-b manifest dark antimicrobial activity “which was unaffected by irradiation of light”. Instead, it was found that irradiation decreased proinflammatory cytokines. In the context of aPDT, it would be useful to comment on this more extensively, as the point of treatment with photosensitizers is usually to trigger the antimicrobial activity upon irradiation only.
Page 5-6, beginning of section 4, it is said that compounds 4 and 6 are used as reference, and later on that polycationic conjugate 7 binds well bacteria. There is no comparison here, it would be useful to give a quantitative comparison in binding efficiency between the reference PS 4 and 6, and the poly-L-lysine conjugate.
Page 6, line 218, what is TAT? The abbreviation is not defined, and no information is given on this class of cell-penetrating peptides. Also, because of this cell-penetrating nature, I think it would be important to comment on the affinity of this nanosystem for healthy mammalian cells compared to pathogenic bacteria.
Page 6-7, lines 231 to 240 and further, unless I’m mistaken there is no reference for this work. The next references cited are in a grouped citation on line 263 (refs 41-43) and are not related to the compounds presented by the authors here. Where are the results on compounds 11-16 cited from? Likewise, lines 264-267, the authors state that “studies have shown” a high efficiency against biofilms, but refer to Chem. Rev. and Chem. Soc. Rev. reviews (refs 44, 45). This is improper referencing, and provides poor access to the information given in all the section, please modify and clarify this.
Page 7, line 240, is there a reason why conjugates 11 and 12 did not affect gram-negative bacteria?
Page 7, line 270, the authors claims that “a considerable fraction of light is absorbed by endogenous chromophores” at 700 nm to justify shifting the absorption of the PS further in the IR. This is slightly misleading as 700 nm is commonly accepted to be part of the first biological transparency window, and the absorption of several endogeneous chromophores (Hb, water, fat…) actually increases slightly towards 750-800 nm compared to 700 nm. However, it would be accurate to stress that the scattering coefficient decreases which improves tissue penetration, and that bacteriochlorins such as compound 3 generally absorb with higher extinction coefficients compared to chlorin analogues.
Page 7, line 285-296, the authors give a lot of experimental details about their own work. I fail to see how this serves the purpose of the results presented here, and of the review in general.
Page 8, lines 297-307, this paragraph is once again not referenced. In addition, to support the discussion, it would be useful to compare the results of compound 16 with monocationic analogue 13.
Page 8, the conclusion is very short, general, and does not even mention chlorophyll derivatives or any of the progress presented in the previous sections at all. Language like “apparently” should also be avoided. This whole conclusion should be reconsidered and revised.
Other minor issues:
Page 1 line 28, the authors write “combined with visible light”. This is not always the case, many photosensitizers are active with UV or IR light (e.g. perinaphthenones in the UV, halogenocyanines, and even the bacteriochlorins described later on in the paper in the near IR).
Page2-3, I feel like the 9-point list loses the reader easily. It would have more impact with increased clarity (different format, rephrasing or an additional figure to support it for example).
Figure 1, y-axis numbers decimals should be “.” Not “,”.
Line 90-92, the sentence is unclear, please rephrase.
Page 3, line 133, this sentence is too vague. Remove or modify.
Line 202, “well binds both with” is incorrect, rephrase into “binds well with both”.
Page 5, line 195, APDT is not defined. Please check abbreviations.
Figure 3, the methyl group on the chlorin structures are sometimes labelled with a “CH3”, sometimes not, while the methyl esters are labelled with a “Me”. Please homogenise this.
Overall, the manuscript needs editing for typographical errors, incorrect punctutation, and grammar, and for formatting issues in references (e.g. 29, 30). A few typos were found in the paper. Line 113, “efflux pump” (no s). Lines 171 199, 248, 266, 268, 302, 313, spaces missing. Figure 2, “derivatives” in the caption. Line 252, “cm2”. Line 265, the N in the IUPAC name of compound must be in italic. Line 302, bacteria should not be in italic. Bacteria names are sometimes written in full, sometimes abbreviated, and not always in italic. Please homogenise.

Author Response
The team of authors would like to thank the referee for the in-depth analysis of the peer-reviewed work and the comments made, which helped to significantly improve the quality of the article.
The articles recommended by the referee were discussed in the text of this paper (lines 117-134, 193-250, 306-349) . According to the comments of the referee, the following work was carried out:
1) Page 1, lines 37-42, the authors write “Photosensitizers are capable of absorbing radiation in the entire visible region of the spectrum”. This is misleading at best, inaccurate at worst. Photosensitizers will absorb light in their absorbance peaks, shining light of other wavelengths cannot result in promotion to the excited state and generation of reactive oxygen species. It might be a shortcut due to the fact that chlorophyll derivatives do have several absorption peaks across the visible, but it remains untrue as general information. Please rectify.
This sentence has been reformulated. Additional information has been added regarding the spectral properties of photosensitizers (lines 37-49).
2) Page 2, lines 51-54, the authors write “it was shown in the 1990s that the susceptibility of gram-positive and gram-negative bacteria to PDT differs.” This sentence is not referenced; the next referenced cited is from 2002 and deals with porphycene–polylysine conjugates for both types of bacteria.
The necessary references have been added (line 59).
3) Page 2, line 69-71, since the authors are talking about a “passive targeting” based on electrostatic interactions between cationic PS and pathogens, it would be beneficial to discuss briefly about the selectivity of this strategy between pathogens and mammalian cells.
A brief explanation of the selectivity of cationic PSs to pathogens has been added (line 75).
4) Page 2, line 72-74, the paragraph is under-referenced.
The necessary references have been added (line 83).
5) Page 2, line 75-80, this paragraph is not referenced, references 12 and 13 should be cited earlier.
References 12 and 13 (new numbers are 24 and 25) have been moved above (line 89) .
6) Page 2, line 82, “many times higher” is vague, giving an order of magnitude would benefit the discussion.
This phrase has been corrected (line 91).
7) Page 3, line 123-124, the authors write “The main difference between photodynamic therapy and antibiotic therapy is the lack of resistance to singlet oxygen”. Although the difference is true, the sentence lacks context, as photodynamic therapy requires the use of light and is therefore a localised treatment, while antibiotic therapy can be applied in the whole body. Reformulation is advised.
The information provided has been reformulated and supplemented (line 136).
8) Page 3, line 138, it would be beneficial to the reader to explain why the reduced pyrrole ring is responsible for a near-infrared absorption with high extinction coefficient in chlorophyll derivatives. Same goes for bacteriochlorophyll derivative 3 in which the effect is even more pronounced. As a more general comment, this section would benefit from having a brief introduction to the different classes of tetrapyrrole macrocycles encountered in the manuscript (porphyrin, chlorin, bacteriochlorin) with comments on the effect of hydrogenation/symmetry on their absorption properties. This would also be the opportunity to discuss a point that is entirely absent from the paper, i.e. the stability of bacteriochlorin derivatives.
The paragraph was supplemented with the necessary information (line 167).
9) Page 4, line 152, the sentence “Today these compounds are widely used in antitumor PDT.” is not referenced.
The necessary references have been added (line 176).
10) Page 4, Figure 1, the axes on the absorption spectrum do not have any title or units.
Page 5, compounds 5c-e by Ayaz et al., shown on figure 2, are not discussed in the paragraph (lines 182-192). In addition, the paragraph focuses on the fact that compounds 5a-b manifest dark antimicrobial activity “which was unaffected by irradiation of light”. Instead, it was found that irradiation decreased proinflammatory cytokines. In the context of aPDT, it would be useful to comment on this more extensively, as the point of treatment with photosensitizers is usually to trigger the antimicrobial activity upon irradiation only.
Figure 1 has been improved. Discussion of the work of Ayaz et al. has been expanded and reformulated (lines 251-267).
11) Page 5-6, beginning of section 4, it is said that compounds 4 and 6 are used as reference, and later on that polycationic conjugate 7 binds well bacteria. There is no comparison here, it would be useful to give a quantitative comparison in binding efficiency between the reference PS 4 and 6, and the poly-L- lysine conjugate.
Reference information has been added (line 279).
12) Page 6, line 218, what is TAT? The abbreviation is not defined, and no information is given on this class of cell-penetrating peptides. Also, because of this cell-penetrating nature, I think it would be important to comment on the affinity of this nanosystem for healthy mammalian cells compared to pathogenic bacteria.
More information about this peptide is provided (line 295).
13) Page 6-7, lines 231 to 240 and further, unless I’m mistaken there is no reference for this work. The next references cited are in a grouped citation on line 263 (refs 41-43) and are not related to the compounds presented by the authors here. Where are the results on compounds 11-16 cited from? Likewise, lines 264-267, the authors state that “studies have shown” a high efficiency against biofilms, but refer to Chem. Rev. and Chem. Soc. Rev. reviews (refs 44, 45). This is improper referencing, and provides poor access to the information given in all the section, please modify and clarify this.
Errors have been fixed. The necessary references have been added.
14) Page 7, line 240, is there a reason why conjugates 11 and 12 did not affect gram-negative bacteria?
Unfortunately, this fact was not investigated in the reviewed work.
15) Page 7, line 270, the authors claims that “a considerable fraction of light is absorbed by endogenous chromophores” at 700 nm to justify shifting the absorption of the PS further in the IR. This is slightly misleading as 700 nm is commonly accepted to be part of the first biological transparency window, and the absorption of several endogeneous chromophores (Hb, water, fat...) actually increases slightly towards 750-800 nm compared to 700 nm. However, it would be accurate to stress that the scattering coefficient decreases which improves tissue penetration, and that bacteriochlorins such as compound 3 generally absorb with higher extinction coefficients compared to chlorin analogues.
This sentence has been reformulated (line 389).
16) Page 7, line 285-296, the authors give a lot of experimental details about their own work. I fail to see how this serves the purpose of the results presented here, and of the review in general.
Excessive experimental information has been removed.
17) Page 8, lines 297-307, this paragraph is once again not referenced. In addition, to support the discussion, it would be useful to compare the results of compound 16 with monocationic analogue 13.
The necessary reference has been added. A short commentary on the comparison of the biological activity of compounds 13 and 16 (new numbers 21 and 18) has been added (line 414)
18) Page 8, the conclusion is very short, general, and does not even mention chlorophyll derivatives or any of the progress presented in the previous sections at all. Language like “apparently” should also be avoided. This whole conclusion should be reconsidered and revised.
The conclusion has been rewritten.
Minor issues:
1) Page 1 line 28, the authors write “combined with visible light”. This is not always the case, many photosensitizers are active with UV or IR light (e.g. perinaphthenones in the UV, halogenocyanines, and even the bacteriochlorins described later on in the paper in the near IR).
Fixed (line 28)
2) Page2-3, I feel like the 9-point list loses the reader easily. It would have more impact with increased clarity (different format, rephrasing or an additional figure to support it for example).
In our opinion, the presentation in the form of a list, on the contrary, is more convenient. However, the text has been abbreviated for ease of reading (lines 100-116).
3) Figure 1, y-axis numbers decimals should be “.” Not “,”.
Fixed
4) Line 90-92, the sentence is unclear, please rephrase.
This sentence has been rephrased (lines 99-101).
5) Page 3, line 133, this sentence is too vague. Remove or modify.
This sentence has been removed.
6) Line 202, “well binds both with” is incorrect, rephrase into “binds well with both”.
Fixed
7) Page 5, line 195, APDT is not defined. Please check abbreviations.
Abbreviation has been corrected
8) Figure 3, the methyl group on the chlorin structures are sometimes labelled with a “CH3”, sometimes not, while the methyl esters are labelled with a “Me”. Please homogenise this.
The label was brought to a single format.
9) Overall, the manuscript needs editing for typographical errors, incorrect punctutation, and grammar, and for formatting issues in references (e.g. 29, 30). A few typos were found in the paper. Line 113, “efflux pump” (no s). Lines 171, 199, 248, 266, 268, 302, 313, spaces missing. Figure 2, “derivatives” in the caption. Line 252, “cm2”. Line 265, the N in the IUPAC name of compound must be in italic. Line 302, bacteria should not be in italic. Bacteria names are sometimes written in full, sometimes abbreviated, and not always in italic. Please homogenise.
All these errors have beed fixed.
Round 2
Reviewer 2 Report
The authors addressed a large part of the comments of the reviewers. A few references suggested in the first report are still missing, and the paper fails to address a certain number of points one could expect from a review on chlorophyll derivatives for aPDT, but the paper is now better referenced globally and of slightly improved quality. However, in the process of revising, the paper has become more difficult to read, and still lacks a clear structure and direction, which fails to give a proper take-home message.
In this sense, the manuscript still needs some English editing (partly because of the revised paragraphs), as follows:
- line 90 to 95, the paragraph is now less clear than before, some rephrasing would help
- line 102, "the direct contact", "the" should be removed (many other improper occurrences of "the", lines 140, 148, 155, 156, 158, 187, 197, 204, 235, 250, 254, 255, 278, 327, 334, 335, 349, 513, 523, 526, 532... making the reading difficult)
- Line 175 "therapy schemes, and mode" is unclear, please rephrase.
- Line 251 "in the chlorophyll a insertion" is unclear, please rephrase.
- Lines 277-280, the list of symptoms is unclear
- Line 291-293, the sentence is unclear and has no verb.
- Line 371 "in the article" is poorly phrased
- Line 381-385, "as" is not adapted to the presentation of the list, and makes the sentence confusing. Please rephrase
- Line 390 increase/enhance permeability is repeated
- Line 396-401, repeated use of the IUPAC names make the paragraph hard to read.
- Line 531, Micobacterium should be Mycobacterium
- Line 533, "their manifestation of pronounced side and general toxic properties" is unclear, should be rephrased
On the content of the manuscript, the following points should be further addressed:
- Line 158, PDZ and FTC are not exactly structural derivatives of chlorin e6, but use specific counter-ions on the carboxylic acids of chlorin e6. The author's formulation is a bit misleading. Also, the whole paragraph (152-172) would be more suited in a place where chlorin e6 is discussed (such as section 4, after mention of compound 5)
- Line 237 Figure 2 is mentioned, but Photolon is not on the figure. If this is cited to refer to the structure of chlorin e6, it is a bit confusing.
- Line 257-258 "singlet oxygen yield" should be quantum yield
- Line 265, PDI is not defined
- On Figure 2, compounds 4g-k are discussed in a completely different place to 4a-f. Why is that?
- Still on Figure 2, chlorin e6 is now numbered 5 and discussed after 4a-f. However chlorin e6 is mentioned and discussed beforehand in the text. This should be fixed. Overall this section is lacking proper structure and direction, and is the weak point of this version of the manuscript.
- As a minor point, chlorin e6 can be considered an anionic photosensitizer because of its carboxylic acids that are deprotonated at physiological pH; as is the case in PDZ and FTC as well. Maybe "anionic" should be added to the title of section 3
- Line 286-290 is a repetition
- Line 357, TAT is defined after the abbreviation has been used
- Line 389 (and 403), do the authors mean TWEEN 80?
- Line 389, EDTA is not defined
- Line 394, the authors state that the pigment is effective in the absence of light. A comment should be added on the dark toxicity, or on the fact that aPDT is usually activated only by light. A similar comment was made in the previous report in another instance.
- Line 403, "solubilizer" should be "surfactant"
- Line 413, are refs 7 and 8 accurate here?
Author Response
Replies to the reviewer after the 2nd review
The authors addressed a large part of the comments of the reviewers. A few references suggested in the first report are still missing, and the paper fails to address a certain number of points one could expect from a review on chlorophyll derivatives for aPDT, but the paper is now better referenced globally and of slightly improved quality. However, in the process of revising, the paper has become more difficult to read, and still lacks a clear structure and direction, which fails to give a proper take-home message.
We again thank the reviewer for the work done. We have included the necessary references that we missed after the first review (line 128), though, we think it was some misunderstanding in the position of the lines (some of the remarks were from 1st version of the article ). Also, the structure of one of the sections of the article has been redone for the convenience of the reader.
Our manuscript was corrected for grammar mistakes by a professional linguist.
Notes on the English language:
1) line 90 to 95, the paragraph is now less clear than before, some rephrasing would help
Rephrased (line 84-86)
2) line 102, "the direct contact", "the" should be removed (many other improper occurrences of "the", lines 140, 148, 155, 156, 158, 187, 197, 204, 235, 250, 254, 255, 278, 327, 334, 335, 349, 513, 523, 526, 532... making the reading difficult)
“The” were removed
3) Line 175 "therapy schemes, and mode" is unclear, please rephrase.
Rephrased (line 112-113)
4) Line 251 "in the chlorophyll a insertion" is unclear, please rephrase.
Rephrased (line 174)
5) Lines 277-280, the list of symptoms is unclear
The list has been deleted (line 247)
7) Line 291-293, the sentence is unclear and has no verb.
Fixed (line 253-154)
8) Line 371 "in the article" is poorly phrased
Rephrased (line 294)
8) Line 381-385, "as" is not adapted to the presentation of the list, and makes the sentence confusing. Please rephrase
Rephrased (lines 303-307)
9) Line 390 increase/enhance permeability is repeated
Rephrased (line 312)
10) Line 396-401, repeated use of the IUPAC names make the paragraph hard to read.
This sentence has been rewritten (lines 320-321)
11) Line 531, Micobacterium should be Mycobacterium
Fixed (line 419)
12) Line 533, "their manifestation of pronounced side and general toxic properties" is unclear, should be rephrased
Rephrased (lines 420-421)
Notes on the content of the article:
1) Line 158, PDZ and FTC are not exactly structural derivatives of chlorin e6, but use specific counter-ions on the carboxylic acids of chlorin e6. The author's formulation is a bit misleading. Also, the whole paragraph (152-172) would be more suited in a place where chlorin e6 is discussed (such as section 4, after mention of compound 5)
The formulation has been clarified. The paragraph itself has been moved to a more appropriate place (lines 224-228).
2) Line 237 Figure 2 is mentioned, but Photolon is not on the figure. If this is cited to refer to the structure of chlorin e6, it is a bit confusing
The sentence has been clarified (line 210-212).
3) Line 257-258 "singlet oxygen yield" should be quantum yield
Fixed (line 180-181).
4) Line 265, PDI is not defined
The abbreviation has been defined (line 189)
5) On Figure 2, compounds 4g-k are discussed in a completely different place to 4a-f. Why is that?
Still on Figure 2, chlorin e6 is now numbered 5 and discussed after 4a-f. However chlorin e6 is mentioned and discussed beforehand in the text. This should be fixed. Overall this section is lacking proper structure and direction, and is the weak point of this version of the manuscript.
As a minor point, chlorin e6 can be considered an anionic photosensitizer because of its carboxylic acids that are deprotonated at physiological pH; as is the case in PDZ and FTC as well. Maybe "anionic" should be added to the title of section 3
The text of this section has been structured. The section title was changed according to the recommendation of the reviewer (line 126).
6) Line 286-290 is a repetition
Rephrased (line 247-253)
7) Line 357, TAT is defined after the abbreviation has been used
Fixed (line 282-283)
8) Line 389 (and 403), do the authors mean TWEEN 80?
Yes, exactly. Fixed. (line 311 and 324).
9) Line 389, EDTA is not defined
Fixed (line 311).
10) Line 394, the authors state that the pigment is effective in the absence of light. A comment should be added on the dark toxicity, or on the fact that aPDT is usually activated only by light. A similar comment was made in the previous report in another instance.
Comment has been added (lines 316-318)
11) Line 403, "solubilizer" should be "surfactant"
Fixed (line 323)
12) Line 413, are refs 7 and 8 accurate here?
This error has been fixed. (333)

Round 3
Reviewer 2 Report
I am pleased to see that the comments from previous review rounds have been taken into account, grammar has been well corrected, and the manuscript has now been substantially improved.
Other than a few typos and formatting issues that can be double checked at the proofreading step (e.g. lines 181 and other occurrences, "quantum singlet oxygen yield" is more commonly named "singlet oxygen quantum yield"; line278 P. aeruginosa should be in italic etc.), the manuscript is now acceptable for publication.